# Effects of Hatch Window and Nutrient Access in the Hatcher on Performance and Processing Yields of Broilers Reared with Equal Hatch Window Representation

**DOI:** 10.3390/ani11051228

**Published:** 2021-04-23

**Authors:** Joshua R. Deines, F. Dustan Clark, Doug E. Yoho, R. Keith Bramwell, Samuel J. Rochell

**Affiliations:** 1Department of Poultry Science, University of Arkansas, Fayetteville, AR 72701, USA; joshuadeines@gmail.com (J.R.D.); fdclark@uark.edu (F.D.C.); 2Jamesway Incubator Company Inc., Cambridge, ON N1R 7L3, Canada; keith.bramwell@jamesway.com

**Keywords:** hatch window, hatcher, feed access, broiler

## Abstract

**Simple Summary:**

In commercial hatcheries, hatched broiler chicks remain in the hatcher without feed or water until all chicks are pulled from the hatching cabinet. Subsequent hatchery holding and transport periods can further delay nutrient access, potentially causing dehydration and limiting the bird’s growth potential. This can possibly be mitigated by providing feed and water in the hatching cabinet to promote immediate nutrient access to chicks after hatching. In the current experiment, chicks hatched in baskets modified to supply feed and water were compared with those hatched in standard baskets in regards to their organ weights, growth performance to 42 d, and processing yields. Additionally, chicks were identified according to moment of hatch within the hatch window to evaluate this factor and its potential interaction with nutrient access. Chicks from hatching baskets with nutrient and water access had heavier body weights during the first 4 wk of growth, but were otherwise similar to chicks from standard baskets in growth performance and meat yield, regardless of the timing of their hatch. This indicates that broilers may be able to compensate for some degree of delayed feed and water access associated with their timing of hatch and subsequent holding and transport.

**Abstract:**

The objective of this experiment was to investigate the effects of feed and water availability in hatching baskets on broiler performance, processing yield, and organ weights while considering the influence of hatch window. Cobb 500 eggs were transferred into illuminated hatchers with two hatching basket types [control (CTL) hatching baskets with no nutrients provided or baskets containing feed and water (FAW)]. Chicks were pulled sequentially to establish four hatch window periods (HWP): early, pre-peak, post-peak, or late. Chicks were then held for 4 h at the hatchery without nutrient access and subsequently reared in 26 floor pens designated as CTL (n = 13) or FAW (n = 13), with 13 chicks from each of the 4 HWP per pen (52 chicks per pen). At 43 d, 16 males from each pen were processed. Chicks from FAW baskets were 1 g heavier (*p* < 0.001) than those from CTL baskets at placement and were heavier through 28 d (*p* = 0.003) but similar (*p* > 0.05) in body weight (BW) for the remainder of the 42 d. No differences (*p* > 0.05) in feed conversion ratio, mortality, or processing data were observed between CTL and FAW groups. Early-hatching chicks were lighter (*p* < 0.001) than those from all other HWP at placement, but were only lighter (*p* < 0.001) than the post-peak group by 42 d. In summary, it was found that hatching basket nutrient access increased the BW of broilers during the first 4 wk of growth, with no other effects on performance or yield. Also, earlier-hatching chicks were generally able to compensate for a lighter placement BW.

## 1. Introduction

The modern broiler has been selected for rapid growth, allowing a desired market weight to be reached in a shorter amount of time compared with broilers from the past [1]. This growth is initiated once the chick begins consuming exogenous feed, which it efficiently converts to body tissue. Feed intake of newly hatched chicks has been shown to increase yolk secretions to the small intestine and increase glucose and methionine uptake [2]. Further, Noy and Sklan [3] reported that sawdust fed to chicks after hatch resulted in a higher BW at 4 d post-hatch compared to a control, indicating that non-nutritive material present in the gastrointestinal tract may also initiate a similar physiological response. As the chick transitions from endogenous to exogenous nutrient sources, the digestive system undergoes significant alterations as it matures with age, and providing feed to chicks soon after hatch increases the rate of this development. Conversely, chick fasting and delayed placement studies have reported depressions in BW that persist to market age, decreased uniformity, and higher mortality [4,5,6]. Prior to water and feed access, chicks are relying on their remaining yolk sac to provide energy [7] and nutrients for growth, and yolk sac degradation accounts for the majority of BW loss during the first 24 h after hatch [4].

In addition to processing at the hatchery and transportation to the farm, the time at which a chick hatches determines how long it has until its first access to water and feed. The hatch window is generally 24 to 48 h for commercially produced broilers and therefore earlier hatching chicks have considerably more time out of the egg until placement. While chicks are held in the hatcher without water and feed, they rapidly begin to decrease in BW as a result of dehydration and yolk utilization [8,9].

Early feeding methods such as in ovo and hatcher feeding have been investigated as strategies to provide immediate access to nutrients and avoid the negative effects of delayed placement [3,10,11,12]. Feeding in hatching baskets is a method that allows chicks access to nutrients immediately after hatching and becoming ambulatory, and Sklan et al. [8] showed that this is an effective approach for decreasing the amount of BW chicks and poults lose while in the hatcher, particularly for earlier hatchers. Similarly, Hollemans et al. [12] demonstrated that chicks provided nutrients in their hatching baskets have higher BW, up to 28 d, than chicks without. However, compensatory growth of fasted and early hatching chicks may diminish these early benefits depending on the final slaughtering age of the birds [13,14]. 

Clearly, the differing degrees of success reported for various early feeding strategies warrant additional investigation. Further, existing literature on early post-hatch feeding has focused on comparing fed birds to those exposed to fasting periods of 24 h or more after pull of hatch, but such extended fasting periods are not typically observed in practical broiler production. With the increasing availability of commercial systems that facilitate nutrient access in the hatcher, the potential benefits of this specific early feeding strategy need to be evaluated. Under these systems, the time at which a chick emerges from the shell within the hatch window will dictate their opportunity for nutrient consumption before pull and influence the potential for water access to offset dehydration. This may affect both weight and physiological status at placement as well as subsequent feeding behavior. As such, the current experiment was conducted to examine the effects of providing nutrient access in the hatching baskets on the subsequent live performance and processing yields of chicks subjected to industry-relevant holding times and reared with equal hatch window representation.

## 2. Materials and Methods

### 2.1. Egg Source and Incubation

Fertile eggs from a single, 36-wk-old Cobb MV × 500 breeder flock were sourced from a commercial broiler integrator. Eggs were transported to the University of Arkansas Poultry Research Farm hatchery and held overnight in an egg storage room at 18.3 °C. A total of 2520 eggs were then set in a single machine (Ps500 Jamesway Incubator Company, Inc., Cambridge, ON, Canada) and incubated using a common broiler profile of 37.6 °C at 55% relative humidity (29.4 °C wet bulb) from 0 to 18 d. Eggs were turned every hour.

### 2.2. Nutrient Access and Hatch Window

At 18 d of incubation, eggs were transferred to the hatcher (Ps500 Jamesway Incubator Company, Inc. Cambridge, ON, Canada) set at 36.7 °C and 54% relative humidity (27.8 °C wet bulb) until time of pull at 504 h. The hatcher was equipped with LED lighting to provide approximately 21.5 lux to the interior in the hatch baskets. Incubated eggs were randomly assigned to 15 control (CTL) hatching baskets with no nutrients provided and 15 baskets containing feed and water (FAW). Standard hatch baskets were modified by creating a center divider. In each basket, 84 eggs were placed on one side, and the other side was devoid of eggs and configured to provide the CTL or FAW (Figure 1) experimental treatment. After hatching, chicks were then moved to the experimental side of the basket as described in Figure 1. Feed and water were provided ad libitum in 50 mL reagent reservoirs (89094-680 VWR International, West Chester, PA, USA) with each FAW basket containing 3 reservoirs of water and 3 reservoirs of feed. The feed provided in FAW baskets was the same crumbled starter subsequently fed to all chicks at placement (Table 1).

In addition to nutrient access in the hatching basket, a sub-plot treatment factor included hatch window period (HWP). A pilot trial was conducted a week in advance by hatching eggs from the same breeder flock to determine the spread of hatch and define a hatch window. Based on these data, it was determined that the hatch window be divided into four HWP based upon number of h before pull at 504 h: early (30 to 24 h), pre-peak (24 to 18 h), post-peak (18 to 12 h), and late (12 to 6 h). Each HWP represented approximately 25% of the total hatch time. At the end of each HWP, all hatched chicks were removed from the egg side of the basket, tagged for identification, and placed into the experimental side of the basket. A chick was defined as hatched once fully and independently cleared from the shell. The HWP in which a chick hatched established the amount of time chicks had access to the experimental hatching baskets (CTL or FAW). Chicks in early, pre-peak, post-peak, and late HWP had 24 h, 18 h, 12 h, of 6 h access to experimental hatching baskets, respectively. No chicks hatched more than 30 h before pull, and the relatively few chicks that hatched within 6 h of pull were not used in this experiment. Hatchability was not measured as a function of basket treatment, but overall hatchability was 87%.

### 2.3. Growout, Sampling, and Processing

At 504 h, experimental baskets were removed from the hatcher. Chicks were segregated by treatment combination. Subsequently, chicks of the same basket treatment (CTL or FAW) were randomly selected by HWP and placed into a chick box designated as CTL or FAW (13 boxes per basket type) to achieve an equal distribution of 14 chicks from each of the 4 HWP (56 chicks total per box). Chicks were held at the hatchery in these boxes for 4 h to simulate commercially relevant processing and holding times, and no nutrients were provided for any chicks during this time. At the conclusion of the holding time, 4 chicks from each box (1 per HWP) were randomly selected and euthanized for yolk and liver sampling. The other chicks were transported to an experimental rearing facility for placement.

All 52 remaining chicks from each box were placed in a single floor pen to assess live performance during a 42 d experiment. The pens measured 1.52 × 3.05 m and contained fresh pine shavings. Each pen had 2 hanging feeders with commercial feed pans and a single water line with 10 nipples. There were 13 replicate pens of FAW and CTL nutrient access treatments, with each pen containing 13 chicks from each of the 4 HWP. At 3 d post-hatch, 4 chicks from each pen (1 per HWP) in each of the 26 pens (13 CTL and 13 FAW) were again randomly selected and euthanized for yolk sacs and liver sampling. Bird weights and feed consumption were recorded weekly by pen. Individual bird weights were taken at 21 d and 42 d to assess uniformity. Mortality and associated weights were recorded daily. Common starter (0 to 14 d), grower (14 to 28 d), and finisher (28 to 42 d) feeds (Table 1) and water were provided ad libitum. Feed was removed from the pens 10 h prior to processing on 43 d. A total of 366 birds were processed using males from each HWP from each pen which were randomly selected, wing-banded, and processed for determination of carcass and parts weights and yields. Following evisceration, birds were chilled in ice water for 4 h before deboning.

### 2.4. Statistical Analysis

The experiment consisted of 8 treatments in a factorial arrangement of 2 hatching basket types (CTL or FAW) × 4 HWP (early, pre-peak, post-peak, or late) in a split-plot design. The whole-plot factor was experimental hatching basket type (n = 13 for main effect), and the sub-plot factor was the HWP (n = 26 for main effect). As such, there were 13 replicate group of each hatching basket and HWP combination. All data were analyzed using SAS 9.4 (Cary, NC, USA). The MIXED procedure was used to conduct a two-way ANOVA, with a Tukey’s multiple comparison test used to separate statistically different means. Any *p* ≤ 0.05 was considered statistically significant.

## 3. Results

### 3.1. Organ Weights 

There was a HWP × basket type interaction (*p* = 0.026) on yolk-free BW at placement (Table 2). This interaction was because chicks in the FAW-early HWP group were 3.83 g heavier (*p* < 0.05) than the CTL basket chicks in the early HWP, but other HWP combinations were similar (*p* > 0.05). There were no main or interactive effects (*p* > 0.05) of basket type on absolute or relative liver or yolk weights at placement. For HWP, absolute and relative yolk weights followed the same general trend, with the highest values for the late hatchers, the lowest values for the early hatchers, and the intermediate values for the pre- and post-peak hatchers. The late HWP group also had the highest (*p* = 0.002) relative yolk weight, 12.85%. The liver weights of day-old chicks had an inverse trend. Chicks from the early HWP had higher (*p* = 0.009) liver weights averaging 1.50 g compared to 1.22 g of chicks from the late HWP; other HWP showed intermediate liver weights (Table 2). The relative liver weight was also highest (*p* = 0.002) for early-HWP chicks at 3.53 %.

At d 3, yolks sampled were no longer significantly different (*p* > 0.05) in absolute or relative weight among treatment groups (Table 3). There was, however, a significant difference in yolk-free BW between HWP (*p* = 0.040) and nutrient access treatments (*p* = 0.010). Liver samples at 3 d post-hatch no longer indicated a difference (*p* > 0.05) in relative liver weights between HWP; however, early-HWP chicks still had a higher (*p* = 0.022) absolute liver weight, 5.26 g, compared to late-HWP chicks, 4.56 g; other HWP were showed intermediate liver weights (Table 3). No absolute or relative liver differences were observed between basket type treatments.

### 3.2. Growth Performance 

At placement, chicks from FAW baskets were 1.0 g heavier (*p* < 0.001) than chicks from CTL baskets. The HWP also affected (*p* < 0.001) placement BW, with the later hatching chicks being the heaviest, but becoming the lightest at 7 d post-hatch (*p* < 0.001; Table 4). The effects of HWP on BW were apparent (*p* ≤ 0.05) throughout the remainder of the experiment but varied in ranking and magnitude. Chicks provided nutrient access in the hatching baskets continued to have higher BW than CTL chicks at 7 d (*p* < 0.001), 14 d (*p* < 0.001), 21 d (*p* < 0.001), and 28 d (*p* = 0.003). The remainder of the 42 d grow-out showed no BW differences (*p* > 0.05) between basket types. During the grow-out period, no HWP × basket type interactions (*p* > 0.05) for BW were observed. Feed consumption was measured by whole pen due to the split-plot arrangement and co-rearing of chicks from different HWP. The higher BW of FAW birds up to d 28 coincided with higher BWG (*p* = 0.009) to 28 d and higher (*p* = 0.005) FI to 21 d, after which, there were no differences (*p* > 0.05; Table 5). Pens of FAW and CTL basket treatments had similar (*p* > 0.05) FCR throughout the 42 d experiment.

The average BW coefficient of variation (CV) was 8.72% at 21 d and 11.36% at 42 d, and there were no differences (*p* > 0.05) among treatment groups (data not shown). Mortality was not affected (*p* > 0.05) by any of the treatments, and the cumulative average was 2.57% at 42 d (data not shown).

### 3.3. Processing Weights and Yields 

The ranking differences among HWP observed for the 42 d liver weight were again observed for the live bird weight of those selected for sampling (*p* = 0.033; Table 6); however, hot and chilled carcass weights were not different (*p* > 0.05). Differences among HWP were found for hot fat pad weight (*p* = 0.021) and yield (*p* = 0.049) which were not separated by Tukey’s but trended to increase in weight and yield from the early to the late HWP. A HWP × basket type interaction was observed for tender yield (*p* = 0.002; Table 7) and wing weight (*p* = 0.029). Leg quarter weight (*p* = 0.009) and yield (*p* = 0.015) were also different among HWP. Nutrient access in the hatching basket did not have any significant effects (*p* > 0.05) on processing weights or yield.

## 4. Discussion

The current study was based on having equal hatch window representation in each pen to reflect the effects of hatching basket feeding when chicks hatching from different periods of the hatch window were commingled, as would be the scenario in commercial production. Though the normal distribution of a typical hatch window would not result in equal representation of all HWP for the flock, equally stratifying birds from all HWP into each pen provided a more valid design for statistical analysis in the current experiment. The comparison of 0 d yolk samples showed that HWP affected the amount of absolute and relative yolk weight. Assuming a similar initial yolk weight, the earlier hatching chicks had more time outside of the egg and thus utilized more yolk. The liver weight, on the other hand, was higher for the early-HWP chicks compared to the late-HWP and intermediate for other HWP. This somewhat inverse relationship between yolk and liver size at placement may indicate a mobilization of yolk nutrients and energy to support tissue growth, especially for organs that have a high allometric priority during early growth such as the liver [2], though additional compositional analyses of liver and yolk nutrient profiles would be needed to confirm this. Chicks from FAW baskets had similar yolk weights to CTL chicks; however, they showed a tendency for higher yolk-free BW that corresponded with a higher placement weight. These chicks were likely more hydrated and were observed to have full crops at placement, which also would have added to their total BW. Average absolute and relative yolk sac weights decreased from approximately 5 to 1 g and 11 to 1% from 0 to 3 d of age and indicated that chicks quickly utilized their yolk sacs during this period. At 3 d post-hatch, the ranking of liver weights by HWP was similar to that observed at 0 d. Liver and yolk sac weight differences among hatch times diminished during early growth, which is supported by hatch moment studies by Lamot et al. [13] who found no differences in liver or yolk sac weight at 4 d post-hatch.

During the grow-out, both basket type and HWP independently affected the performance of the broilers, but the effects of each factor varied among time points. Early-hatching birds were lighter at placement than birds from other HWP, but interestingly became heavier than the post-peak and late-HWP chicks at 7 d. There was clearly a faster growth rate in order for this compensation occur, but due to the pen arrangement in which birds from all HWP shared a common feeder, we were unable to identify if this compensation in BW was due to increased FI, better FCR, or both. This is in agreement with studies by Lamot et al. [13] who also found that earlier-hatching chicks were initially lighter, but had accelerated growth after placement as a result of higher FI, which combined with a longer amount of growth time, allowed them to become heavier than their later-hatching counterparts by 4 d post-hatch, though this study ended at 18 d. In the current experiment, the final 42 d BW of the early HWP was again lower that of chicks from post-peak HWP, indicating that the early compensatory gain may not be sustained to market ages, though ranking in BW varied among HWP between 14 and 42 d. Thus, while HWP does seem to influence growth rate, more mechanistic evaluation of these effects are necessary to fully understand the relationship between HWP and growth potential.

Chicks from FAW baskets were heavier than CTL birds at each time point to 28 d. This was likely driven by an early initiation of feed consumption in the hatcher as shown by a 1 g BW advantage at placement, and the higher FI to 21 d supported higher BWG to 28 d. Beyond 28 d, the FAW birds were similar in BW to the birds from CTL baskets. Converging BW of hatch basket fed and control chicks at 28 d of age were also reported by Hollemans et al. [12]. Sklan et al. [8] observed effects of hatch basket feeding in a shorter 21 d experiment during which fed chicks exhibited higher BW for the trial duration. The type of early feeding, time of initiation, and duration of fasting for control chicks and duration of the grow-out have all led to varying degrees of success with approaches to early feeding. For example, Kidd et al. [14] fed chicks in trays during post-hatch, pre-placement holding and did not observe any differences in BW, feed efficiency, or mortality among treatments beyond 7 d of age.

Providing water and feed in hatching baskets does allow immediate access to nutrients once a chick hatches and also initiates feeding and growth at differing times for the same hatch of chicks to be reared together. Therefore, BW CV were considered important to evaluate uniformity among treatments as an indicator of overall broiler health and performance [15]. The CV for all treatment groups and low cumulative mortality indicated this was a well-performing flock. Perhaps, a more challenging environment would have generated noticeable differences among the groups, since the current experiment utilized eggs from a prime-age breeder flock and fresh litter. Further, a longer post-hatch holding period to simulate stress associated with extended transport times may have resulted in a different response to hatching tray feed access than observed in the current experiment.

Live weight differences among HWP at 42 d were similar in relation to the live weights at processing, where the post-peak HWP group was heavier than the early HWP chicks, with other groups having intermediate weights. These weights were overall heavier than BW in Table 4 due to the sub-sample of only males being used for processing. Fat pad weight and yield differences among HWP were also detected, which may reflect differences in body composition and feed efficiency, but this cannot be confirmed due to the common pen rearing of all HWP and a lack of more extensive body composition analyses. Processing data at 42 d surprisingly revealed a HWP × basket type interaction for tender yield that was due to a decrease in tender yield from early- to late-HWP groups for the CTL group, while tender yields for FAW groups were similar across all HWP. An interaction was also observed for wing weight, whereby FAW access tended to increased wing weight for early and pre-peak hatchers but generally decreased wing weight for post-peak and late-hatching chicks. The leg quarter weight and yield were the only processing measurements independently influenced by HWP. The absolute leg quarter weight ranked the same among HWP as the live weights, but interestingly, this was the only part that was statistically separated. This may indicate that differences in dark meat accretion may have been more influenced by HWP than breast muscle accretion.

## 5. Conclusions

In conclusion, results from this experiment indicate that feed and water access in the hatcher may lead to higher chick weights at placement and increase the weight of broilers during the first 28 d of growth. However, feed and water access in the hatcher had no influence on final 42 d body weight, processing yield, FCR, or mortality. These findings also highlighted an intriguing BW compensation of early-hatching chicks, which were lighter at hatch but became heavier than their later-hatching counterparts at 7 d, with varying weight differences at later ages. The current study generally showed merging of weights by basket type and compensation of BW by HWP, suggesting that the BW of broilers at placement or 7 d may not always be an accurate predictor of BW at market. This is important in reviewing similar early post-hatch feeding studies due to the fact that none, to the authors’ knowledge, have processed or grown birds beyond 21 d without fasting for extended periods. Further research using varying holding times and hatch window lengths, in addition to more challenging environmental conditions (e.g., heat stress, transport stress, or sub-clinical disease) would assist in determining if and when hatching basket feeding yields an advantage at market and processing.

## Figures and Tables

**Figure 1 animals-11-01228-f001:**
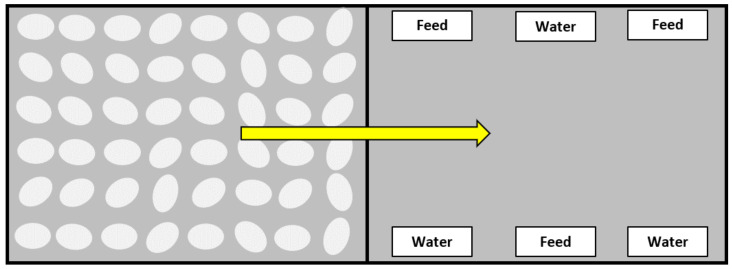
Layout of experimental hatching baskets. A standard hatching basket was divided down the center using mesh wire. At transfer, 84 eggs were placed in one side. At the conclusion of each hatch window period (HWP), hatched chicks were physically moved from the egg side of the basket to the treatment side that was without (control, CTL) or with feed and water (FAW).

**Table 1 animals-11-01228-t001:** Ingredient and calculated nutrient composition of common diets.

Item	Starter (0 to 14 d)	Grower (14 to 28 d)	Finisher (28 to 42 d)
Ingredient composition, %			
Corn	60.98	64.72	69.55
Soybean meal (46.3%)	34.30	30.14	25.15
Poultry fat	1.16	1.87	2.33
Limestone	1.11	1.07	1.02
Dicalcium phosphate	1.05	0.92	0.74
Sodium chloride	0.44	0.44	0.40
DL-methionine	0.32	0.26	0.25
L-lysine	0.21	0.17	0.20
L-threonine	0.09	0.07	0.09
Mineral premix ^1^	0.10	0.10	0.10
Vitamin premix ^2^	0.10	0.10	0.10
Choline chloride (60%)	0.05	0.04	0.04
Selenium premix (0.06%)	0.02	0.02	0.02
Coccidiostat ^3^	0.05	0.05	-
Phytase ^4^	0.03	0.03	0.03
Calculated nutrient composition, % unless otherwise noted			
AME_n_, kcal/kg	3008	3086	3167
CP	21.45	19.65	18.00
Digestible Lysine	1.18	1.05	0.95
Digestible TSAA	0.89	0.80	0.74
Digestible Threonine	0.77	0.69	0.65
Calcium	0.90	0.84	0.76
Available phosphorus	0.45	0.42	0.38

^1^ Supplied the following per kg of diet: vitamin A, 6350.29 IU; vitamin D_3_, 4535.92 ICU; vitamin E, 45.36 IU; vitamin B_12_, 0.01 IU; menadione, 1.24 mg; riboflavin, 5.44 mg; d-pantothenic acid, 8.16 mg; niacin, 31.75 mg; folic acid, 0.73 mg; pyridoxine, 2.27 mg; thiamine, 1.27 mg; biotin, 0.07 mg. ^2^ Supplied the following per kg of diet: calcium, 55.5 mg; manganese, 100 mg; magnesium, 27 mg; zinc, 100 mg; iron, 50 mg; copper, 10 mg; iodine, 1 mg. ^3^ Bio-Cox^®^ 60, Huvepharma, Peachtree City, GA, USA (salinomycin sodium). ^4^ OptiPhos^®^ 2000 PF, Huvepharma, Peachtree City, GA, USA.

**Table 2 animals-11-01228-t002:** Effects of hatch window period (HWP) and nutrient access of feed and water (FAW) or control (CTL) in the hatching basket on body, yolk, and liver weights of day-old broiler chicks.

Item	Yolk-Free BW (g)	Yolk		Liver	
		Weight (g)	Relative to BW (%)	Weight (g)	Relative to BW (%)
Interaction means (n = 13)					
Early—CTL	36.39 ^b^	4.38	10.58	1.34	3.30
Early—FAW	40.22 ^a^	3.98	9.00	1.65	3.75
Pre-peak—CTL	38.23 ^a,b^	5.06	12.30	1.32	2.87
Pre-peak—FAW	38.25 ^a,b^	4.51	10.53	1.32	3.10
Post-peak—CTL	39.06 ^a,b^	4.99	11.22	1.39	3.21
Post-peak—FAW	38.98 ^a,b^	5.51	12.39	1.30	2.92
Late—CTL	37.85 ^a,b^	5.61	12.80	1.21	2.81
Late—FAW	38.21 ^a,b^	5.67	12.98	1.22	2.78
SEM	0.792	0.397	0.852	0.084	0.207
Main effect of HWP (n = 26)					
Early	38.31	4.18 ^b^	9.79 ^b^	1.50 ^a^	3.53 ^a^
Pre-peak	38.24	4.78 ^a,b^	11.41 ^a,b^	1.32 ^a,b^	2.98 ^b^
Post-peak	39.02	5.25 ^a^	11.80 ^a,b^	1.34 ^a,b^	3.06 ^a,b^
Late	38.03	5.64 ^a^	12.85 ^a^	1.22 ^b^	2.80 ^b^
SEM	0.549	0.276	0.590	0.058	0.143
Main effect of basket (n = 13)					
CTL	37.88	5.01	11.72	1.31	3.05
FAW	38.91	4.91	11.20	1.37	3.14
SEM	0.381	0.193	0.410	0.041	0.099
*p*-values					
HWP	0.594	0.002	0.002	0.009	0.002
Basket	0.054	0.726	0.361	0.299	0.519
HWP × basket	0.026	0.505	0.224	0.089	0.276

^a,b^ Means within a column that do not share a common superscript are significantly different (*p* < 0.05) as determined by a Tukey’s multiple comparison test.

**Table 3 animals-11-01228-t003:** Effects of hatch window period (HWP) and nutrient access of feed and water (FAW) or control (CTL) in the hatching basket on body, yolk, and liver weights of 3 d old broiler chicks.

Item	Yolk-Free BW (g)	Yolk		Liver	
		Weight (g)	Relative to BW (%)	Weight (g)	Relative to BW (%)
Interaction means (n = 13)					
Early—CTL	80.62	1.15	1.41	4.83	5.92
Early—FAW	89.07	0.79	0.87	5.69	6.33
Pre-peak—CTL	81.64	0.99	1.20	4.73	5.71
Pre-peak—FAW	84.43	0.96	1.13	4.79	5.61
Post-peak—CTL	80.00	0.98	1.22	4.72	5.80
Post-peak—FAW	84.04	1.02	1.22	5.23	6.11
Late—CTL	79.86	1.03	1.27	4.68	5.79
Late—FAW	78.83	1.23	1.57	4.43	5.54
SEM	1.912	0.139	0.175	0.233	0.228
Main effect of HWP (n = 26)					
Early	84.85 ^a^	0.97	1.14	5.26 ^a^	6.13
Pre-peak	83.03 ^a,b^	0.98	1.16	4.76 ^a,b^	5.66
Post-peak	82.02 ^a,b^	1.00	1.22	4.97 ^a,b^	5.95
Late	79.34 ^b^	1.13	1.42	4.56 ^b^	5.66
SEM	1.352	0.098	0.124	0.165	0.161
Main effect of basket (n = 13)					
CTL	80.53 ^b^	1.04	1.28	4.74	5.80
FAW	84.09 ^a^	1.00	1.20	5.04	5.90
SEM	0.956	0.0694	0.087	0.116	0.114
*p*-values					
HWP	0.040	0.605	0.359	0.022	0.113
Basket	0.010	0.697	0.533	0.076	0.565
HWP × Basket	0.106	0.232	0.126	0.094	0.411

^a,b^ Means within a column that do not share a common superscript are significantly different (*p* < 0.05) as determined by a Tukey’s multiple comparison test.

**Table 4 animals-11-01228-t004:** Body weight (g) of broilers from different hatch window periods (HWP) within a single hatch window provided feed and water (FAW) in the hatching basket compared to a control (CTL) ^1^.

Item	Days of Age						
	0	7	14	21	28	35	42
Interaction means							
Early—CTL	42.3	185	458	918	1610	2235	2810
Early—FAW	43.7	197	479	963	1649	2266	2885
Pre-peak—CTL	43.2	185	456	910	1609	2262	2871
Pre-peak—FAW	44.4	195	475	968	1659	2309	2937
Post-peak—CTL	44.1	181	457	928	1655	2273	2965
Post-peak—FAW	45.1	190	466	950	1665	2326	2951
Late—CTL	44.8	178	446	920	1594	2291	2915
Late—FAW	45.1	182	456	938	1631	2283	2941
SEM	0.25	1.8	4.0	11.4	16.1	33.9	33.8
Main effect of HWP							
Early	43.0 ^c^	191 ^a^	458 ^a^	940	1629 ^a,b^	2250	2847 ^b^
Pre-peak	43.8 ^b^	190 ^a,b^	456 ^a^	939	1634 ^a,b^	2286	2904 ^a,b^
Post-peak	44.6 ^a^	186 ^b^	457 ^a^	939	1660 ^a^	2300	2958 ^a^
Late	45.0 ^a^	180 ^c^	446 ^b^	929	1612 ^b^	2287	2928 ^a,b^
SEM	0.18	1.2	2.8	7.9	11.2	23.5	23.4
Main effect of basket							
CTL	43.6 ^b^	182 ^b^	454 ^b^	919 ^b^	1617 ^b^	2265	2890
FAW	44.6 ^a^	191 ^a^	469 ^a^	955 ^a^	1651 ^a^	2296	2928
SEM	0.124	0.9	2.0	5.7	8.1	17.0	16.9
*p*-values							
HWP	<0.001	<0.001	0.001	0.706	0.028	0.490	0.010
Basket	<0.001	<0.001	<0.001	<0.001	0.003	0.189	0.106
HWP × Basket	0.146	0.101	0.376	0.224	0.637	0.796	0.528

^a–c^ Means within a column that do not share a common superscript are significantly different (*p* < 0.05) as determined by a Tukey’s multiple comparison test. ^1^ Mean values of 13 (0 d) or 12 (7, 14, 21, 38, 35, 42 d) birds, less mortality, from each HWP in each of the 13 replicate pens of FAW or CTL basket treatment.

**Table 5 animals-11-01228-t005:** Live performance of broilers provided feed and water (FAW) nutrient access in the hatching basket compared to a control (CTL) from 0 to 42 d post-hatch ^1^.

Item	CTL	FAW	SEM	*p*-Value
0 to 14 d				
BWG (g)	410	424	2.9	0.002
FI (g)	521	542	4.4	0.002
FCR	1.279	1.287	0.0062	0.414
14 to 28 d				
BWG (g)	BWG (g)	BWG (g)	BWG (g)	BWG (g)
FI (g)	FI (g)	FI (g)	FI (g)	FI (g)
FCR	FCR	FCR	FCR	FCR
0 to 28 d				
BWG (g)	1573	1606	8.4	0.009
FI (g)	2541	2552	35.7	0.826
FCR	1.619	1.592	0.0216	0.388
28 to 42 d				
BWG (g)	1275	1279	18.9	0.867
FI (g)	2722	2730	37.6	0.879
FCR	2.135	2.136	0.0199	0.956
0 to 42 d				
BWG (g)	2848	2886	22.3	0.232
FI (g)	5263	5282	62.8	0.829
FCR	1.837	1.821	0.0140	0.419

^1^ Mean values of 13 replicate pens of FAW or CTL basket treatment.

**Table 6 animals-11-01228-t006:** Processing weight and yield of broilers from different hatch window periods (HWP) within a single hatch window provided feed and water (FAW) in the hatching basket compared to a control (CTL) ^1^.

Item	Live	Hot Carcass		Hot Fat Pad		Chilled Carcass	
	Weight (g)	Weight (g)	Yield (%)	Weight (g)	Yield (%)	Weight (g)	Yield (%)
Interaction means							
Early—CTL	3152	2389	75.74	46.8	1.49	2432	77.14
Early—FAW	3200	2414	75.39	48.0	1.50	2457	76.75
Pre-peak—CTL	3216	2424	75.34	49.1	1.53	2471	76.83
Pre-peak—FAW	3242	2442	75.29	46.4	1.42	2484	76.61
Post-peak—CTL	3296	2478	75.17	50.8	1.55	2525	76.58
Post-peak—FAW	3263	2456	75.24	50.8	1.52	2503	76.70
Late—CTL	3250	2433	74.87	50.8	1.57	2480	76.28
Late—FAW	3192	2390	74.86	52.2	1.64	2431	76.14
SEM	39.2	32.6	0.346	1.88	0.058	33.2	0.362
Main effect of hatch period							
Early	3176 ^b^	2401	75.56	47.4	1.49	2445	76.95
Pre-peak	3229 ^a,b^	2433	75.32	47.8	1.48	2478	76.72
Post-peak	3280 ^a^	2467	75.20	50.8	1.54	2514	76.64
Late	3221 ^a,b^	2411	74.87	51.5	1.60	2455	76.21
SEM	26.1	21.7	0.231	1.25	0.039	22.1	0.242
Main effect of basket							
CTL	3229	2431	75.28	49.4	1.53	2477	76.71
FAW	3224	2425	75.20	49.4	1.52	2469	76.55
SEM	18.1	15.1	0.160	0.86	0.027	15.3	0.167
*p*-values							
HWP	0.033	0.105	0.157	0.021	0.049	0.088	0.140
Basket	0.868	0.783	0.692	0.993	0.685	0.684	0.477
HWP × basket	0.390	0.602	0.917	0.574	0.329	0.583	0.882

^a,b^ Means within a column that do not share a common superscript are significantly different (*p* < 0.05) as determined by a Tukey’s multiple comparison test. ^1^ Mean values of 4 male birds randomly sampled from each HWP within each of 13 replicate pens per basket treatment.

**Table 7 animals-11-01228-t007:** Processing weight and yield of broilers from different hatch window periods (HWP) within a single hatch window provided feed and water (FAW) in the hatch basket compared to a control (CTL) ^1^.

Item	Breast		Tender		Leg Quarter		Wing	
	Weight (g)	Yield (%)	Weight (g)	Yield (%)	Weight (g)	Yield (%)	Weight (g)	Yield (%)
Interaction means								
Early—CTL	656.3	20.78	134.8	4.28 ^a^	703.9	22.34	252.2	8.01
Early—FAW	647.9	20.22	133.6	4.18 ^a,b^	721.1	22.53	261.7	8.18
Pre-peak—CTL	652.4	20.26	134.5	4.18 ^a,b^	737.1	22.93	254.9	7.93
Pre-peak—FAW	668.9	20.58	136.9	4.22 ^a^	733.0	22.59	259.6	8.02
Post-peak—CTL	672.5	20.36	136.9	4.15 ^a,b^	741.6	22.51	266.7	8.10
Post-peak—FAW	667.9	20.41	137.0	4.19 ^a^	734.6	22.48	259.3	7.96
Late—CTL	664.9	20.43	129.7	3.99 ^b^	721.3	22.20	260.0	8.01
Late—FAW	643.9	20.17	135.1	4.23 ^a^	710.6	22.25	254.5	7.98
SEM	12.51	0.244	2.26	0.049	10.53	0.190	3.66	0.079
Main effect of HWP								
Early	652.1	20.50	134.2	4.23	712.5 ^b^	22.43 ^a,b^	257.0	8.09
Pre-peak	660.7	20.42	135.7	4.20	735.1 ^a,b^	22.76 ^a^	257.2	7.98
Post-peak	670.2	20.38	137.0	4.14	738.1 ^a^	22.50 ^a,b^	263.0	8.03
Late	654.4	20.30	132.4	4.11	715.9 ^ab^	22.22 ^b^	257.2	7.99
SEM	8.34	0.163	1.51	0.033	7.02	0.127	2.44	0.052
Main effect of basket								
Control	661.5	20.46	134.0	4.15	726.0	22.50	258.4	8.01
FAW	657.2	20.34	135.7	4.20	724.8	22.46	258.8	8.03
SEM	5.77	0.113	1.04	0.023	4.86	0.088	1.69	0.036
*p*-values								
HWP	0.359	0.844	0.114	0.053	0.009	0.015	0.161	0.364
Basket	0.579	0.455	0.237	0.097	0.859	0.774	0.886	0.648
HWP × basket	0.387	0.216	0.388	0.002	0.476	0.445	0.029	0.134

^a,b^ Means within a column that do not share a common superscript are significantly different (*p* < 0.05) as determined by a Tukey’s multiple comparison test. ^1^ Mean values of 4 male birds randomly sampled from each HWP within each of 13 replicate pens per basket treatment.

## Data Availability

Data are available on request from the corresponding author.

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
