# Peer review of "Effects of Hatch Window and Nutrient Access in the Hatcher on Performance and Processing Yields of Broilers Reared with Equal Hatch Window Representation"

_animals, 2021, doi:10.3390/ani11051228_

Round 1

Reviewer 1 Report

This study investigated the effects of feed and water availability in hatching baskets on broiler performance, processing yield, and organ weights while considering the influence of hatch window. The study is ethically acceptable. However, the experimental design of the study is not clearly presented. I had to admit that I used an excel file in order to calculate the birds per experiment, per treatment and per examined parameter. In addition, the low sampling number in some parameters does not allow statistical analysis (lines 143-145). Furthermore, the results were not analytically discussed and were not sufficiently supported by references. In particular, the results are supported by only 5 references in the discussion section. The authors did not respond to the suggested major comments. Therefore, my decision is “rejected” for publication in Animals.

Reviewer 2 Report

Thanks for responding to reviewer comments. 

Reviewer 3 Report

I am satisified with the reviewers comments, and will accept the revisions for publication.

This manuscript is a resubmission of an earlier submission. The following is a list of the peer review reports and author responses from that submission.

Round 1

Reviewer 1 Report

This study investigated the effects of feed and water availability in hatching baskets on broiler performance, processing yield, and organ weights while considering the influence of the hatch window. The study is ethically acceptable. However, the experimental design of the study is not clearly presented. In addition, the low sampling number in some parameters does not allow statistical analysis. Furthermore, the results were not analytically discussed and were not sufficiently supported by references. Therefore, my decision is “major revision” for publication in Animals.

Minor comments

Line 90

How did you manage to control relative humidity with the adjustment drinkers in the hatchers?

Author Response

Comments and Suggestions for Authors: This study investigated the effects of feed and water availability in hatching baskets on broiler performance, processing yield, and organ weights while considering the influence of the hatch window. The study is ethically acceptable. However, the experimental design of the study is not clearly presented. In addition, the low sampling number in some parameters does not allow statistical analysis. Furthermore, the results were not analytically discussed and were not sufficiently supported by references. Therefore, my decision is “major revision” for publication in Animals.

The authors hope that changes in the revised manuscript and corresponding responses herein have provided more clarity on the experimental design. Regarding the concern for sampling numbers - the authors would like to clarify there were 13 replicate groups of each treatment combination in this split-plot design. Refer to section 2.4 for a revised description of the experimental design and statistical analysis. The authors feel this is sufficient replication based on preliminary power analyses as well as the statistical sensitivity of the final data reflected by the mean differences and standard errors.

Minor comments

Line 90: How did you manage to control relative humidity with the adjustment drinkers in the hatchers?

The humidity was maintained at the desired industry-relevant setpoint by the automatic control of the machine. When the machine needs humidity, it calls for humidification via water line spray nozzles; when the machine has high humidity, the damper opens wider and the fan can exhaust air to bring the humidity down to set the point. As with any hatch, the chicks hatching from the egg are a large contributor of the moisture/humidity within a machine. However, the hatchers were not at capacity and the relatively small amount of water in the machine allowed the hatcher controls to maintain the correct humidity as monitored by an integrated alarm and log system.

Reviewer 2 Report

Well designed and executed research.  I find this interesting in light of other research studying the hatching of eggs in the broiler facility on the floor with special incubators.  This also allows birds immediate access to feed and water after hatch.  It will be good to compare those results with yours. 

I find the results that indicate no real differences at slaughter between controls and feed and water groups.  This demonstrates the physiological ability for compensatory growth after exposure to early stressful situations.  This is an important result of your research.

I would have liked to see a photo showing the placement of the feed and water dishes in the hatching tray and how the birds reacted.

I only found two minor grammar/spelling errors for correction.

Lines 65-66  change "access nutrients"  to "to access nutrients"

Line 258  change "basket" to "baskets"

Author Response

Comments and Suggestions for Authors: Well designed and executed research.  I find this interesting in light of other research studying the hatching of eggs in the broiler facility on the floor with special incubators.  This also allows birds immediate access to feed and water after hatch.  It will be good to compare those results with yours.

The authors appreciate the positive feedback and agree this study has significant industry relevance considering that multiple incubator manufacturers offer feed and water systems within the hatcher. We would also like to see comparisons of in-house hatching and hatcher nutrient access in future research.

I find the results that indicate no real differences at slaughter between controls and feed and water groups.  This demonstrates the physiological ability for compensatory growth after exposure to early stressful situations.  This is an important result of your research.

We agree that this was an interesting demonstration of compensatory growth that is seen in other early feeding approaches. However, the slaughter component is unique to this paper, as very few early feeding studies have addressed birds at market ages.

I would have liked to see a photo showing the placement of the feed and water dishes in the hatching tray and how the birds reacted.

A figure has been added to show this.

I only found two minor grammar/spelling errors for correction.

Lines 65-66  change "access nutrients"  to "to access nutrients"

  • This has been corrected and now reads “access to nutrients”.

Line 258  change "basket" to "baskets"

  • This has been corrected.

Reviewer 3 Report

Please see comments in attached document. Although I find this area immensely interesting and of great importance to the poultry industry, this paper was very superficial in regards to the authors findings and their interpretation. It was very much a practical paper, looking at production outcomes. I would have like to have seen more science, specifically physiological responses around early feeding in hatching baskets rather than just organ and bodyweights. Also the practicality of providing feed and water in hatching baskets should also be addressed.

Author Response

Comments and Suggestions for Authors: Please see comments in attached document. Although I find this area immensely interesting and of great importance to the poultry industry, this paper was very superficial in regards to the authors findings and their interpretation. It was very much a practical paper, looking at production outcomes. I would have like to have seen more science, specifically physiological responses around early feeding in hatching baskets rather than just organ and bodyweights. Also the practicality of providing feed and water in hatching baskets should also be addressed.

  • There are many papers addressing physiological responses to feeding and fasting, but there are very few that address the impacts of hatch basket feeding on production. This paper aimed to investigate the effects of providing nutrients to chicks in hatch baskets on performance, with some indirect indications of physiological change (yolk and liver weight). While we agree that there are many possibilities for more extensive sampling and mechanistic analyses, we believe this comprehensive description of growth outcomes, feed efficiency, and processing measurements in response to hatcher feeding is still a valuable addition to literature.
  • Regarding the practicality of providing feed and water in hatching baskets, several commercially available systems have become available in recent years to accomplish this (additions on this made in the Introduction section). However, to our knowledge, there are currently no published data on the effects of these systems on broiler growth and performance. The authors recognize that there are many management, biosecurity, and economic considerations that should be addressed for these systems, but these are beyond the scope of this paper.

Line 68-69: Depends on what you wanting to achieve here. If you can get more birds to market weight and reduce mortality % through early feeding then you can increase output through the total number of birds, rather than through increasing growth and yield of individuals.

  • This statement has been further qualified. However, in this study mortality was not different among treatments with a cumulative average of 2.57% at 42 d. This is stated in section 3.2, lines 199-201.

Line 72: but you mentioned the hatch window was 24-48 hours?

  • In existing papers cited, fasting generally takes place after the total hatch and pull. In these cases, a bird fasted for 48 h during the hatch window and subsequently fasted for 24 has now been fasted for 72 h and would likely be much more responsive to nutrient provision. This is why we feel the current study, which looks at the presence or absence of nutrients during the hatching, but prior to pull of hatch, is important. Prior research starts the “fasting” timeline at hatch pull and not true hatch time, whereas this study investigates the timeline starting at hatch (considering the hatch window). Text has been added to clarify.

Line 76-79: The rationale is not clear, suggest a slight restructure of the introduction. Describe the advantages and disadvantages of feeding in hatching baskets. What is the "GAP" here?

  • Additional text in the Introduction has been added to clarify. As mentioned above, this study was designed to quantify just what those potential advantages and disadvantages are for live performance and processing. There are many commercially available in hatcher feeding systems (e.g. HatchTech) utilized throughout the world, but no clear data have been published about the impacts. There are potential logistical/management disadvantages (and possible pathogen introduction), but the authors believe those factors are better suited for another study or paper after the baseline impacts to the bird are established.

Line 84: Were the eggs pre-incubated prior to setting?

  • No, the eggs were stored as described and then put directly into the incubator. With the number of eggs used and a dedicated incubator, pre-warming was not necessary as they would warm evenly together in the machine.

Line 90: Seems low for hatching conditions

  • This is normal for hatcher humidity and is a common setting for industry application and our university hatchers. These parameters are described in the Cobb-Vantress Hatchery Guide.

Line 99-101: How were they arranged, how did you ensure the chicks did not knock the reservoirs over? you are assuming all the chicks could access the feed and water, did you observe them using the reservoirs?

  • A figure has been added to show this arrangement. No reservoirs tipped over during the experiment. We chose the reagent reservoirs described because they have a low profile and wide base. These were also tested during a preliminary hatch. Some chicks may have of course stepped/fell in them (no drownings occurred). All chicks had access and without the shell/hatch residue there was ample room to access feed and water. We did generally observe chicks consuming both feed and water, which was supported by crop fill (palpation) and BW at pull.

Table 1: were these commerical diets or did you formulate and make your own?

  • These diets were formulated internally to specifications from the Cobb Broiler Guide (these were Cobb chicks) at our university feed mill. The term “common” in the table title is used to signify that diet was common across treatment groups.

Line 102: New Subheading

  • The authors believe that the previous heading “2.2. Nutrient access and hatch window” applies to this paragraph as well. The previous paragraph discusses nutrient access, followed by discussion of hatch window in the second paragraph. Because these are the interrelated factors of the design, the authors respectfully request to keep as is.

Line 105-108: How did you monitor this? Cameras?

  • Although cameras would be beneficial, they were not utilized. Based upon existing literature, and previous research of the authors, the likely spread of hatch/window was plotted to a timeline. From that, the pilot trial eggs were visually observed by opening the hatcher for a brief period during hatch every 2 hours and number hatched were recorded to create the distribution. The peak of this curve became the bases for the HWP used in this experiment.

Line 111: not dried?

  • Dry versus wet is difficult to objectively determine and would require more time with the hatchers open or handling chicks. Additionally, wet chicks can walk in the baskets and potentially access feed. Separation from shell was chosen to be the repeatable indicator of hatched or not hatched.

Line 122: 1 chick to represent each HWP, how can you account for variability with n=1?

  • There was 1 chick sampled from each HWP from each of the 26 boxes . The total n=13 for each treatment combination of basket and HWP, and n=26 for the main effect of HWP. This has been clarified in section 2.4

Line 123: why yolk sac and liver specifically? I am assuming you also weighed the chick?

  • Yolk utilization is an important when it comes to fasting and the switch from yolk-derived to exogenous nutrient sources. With the access to nutrients we wanted to understand if chicks would utilize more or less yolk compared to control. Liver is often measured in nutrition studies. The liver goes through significant physiological changes after hatch and can most basically be visually observed by the color change as the chicks switch from a primarily lipid (yolk) energy source to carbohydrate (feed) energy source. Prior research has suggested that fasted chicks after hatch will eventually deplete liver glycogen reserves, making gluconeogenesis a necessity (leading to a lower BW).
  • Chicks were weighed and that allowed us to get the yolk-free BW and organ weights relative to BW as presented in Table 2 and 3. The remaining BW are presented in Table 4.

Line 130: Again not enough birds sampled here.

  • Please see the notes from above regarding sampling number.

Line 135: Italics

  • This has been corrected.

Line 137: why only males?

  • With limited processing capacity, we were restricted to the number of birds processed. By choosing only one sex, this eliminated potential bias associated with unequal distribution of sex within the straight-run flock. Males were chosen over females as they are the faster growing sex and differences in weight between treatments may be more detectable.

Line 145: did you check for normality?

  • Data were confirmed for normality based on visual observations of graphical representation of error distribution using a QQ plot.

Line 148: What was your hatch rate? Mortality rates?

  • The treatments were not designed to impact hatch so percentage was not measured per basket, but overall hatch was 87%. In this study mortality was not different among any treatments with a cumulative average of 2.57% at 42 d. This is stated in section 3.2.

Line 151-153: did you use bodyweight as a co-variate in your statistical model?

  • Given that body weights were not independent of treatments even at their first measurement (hatch and placement), it would not have been appropriate to use BW as a covariate.

Table 5: “FCR” males and females?

  • Chicks were not sexed at hatched and reared as straight-run. With males and females in a common pen with a common feeder, FCR could not be determined for each sex independently.

Table 7: Too many tables, only present the significant results  in tables and report the rest in text

  • The authors agree that this manuscript does have many tables but feel all are necessary to accurately and efficiently provide the results to the reader. This table (Table 7) for example could not be eliminated as there are significant differences in tender and leg quarter weights. The remaining parts should be present to show that they were not affected, especially for breast meat, which is a primary focus for many producers.

Line 216: commingled

  • This has been corrected.

Line 220-222: Or they used more yolk trying to hatch out. Energy demand of hatching is huge.

  • Very interesting thought and would be a good study. Hatching does have a significant energy demand. However, the statement we made is based off of knowledge that yolk is utilized over time (decrease in weight) and to our knowledge there is no prior research to suggest that earlier hatching chicks have a more challenging time hatching out (energy utilization) than other chicks within a HW.

Line 224-225: how can you conclude that? Of the liver? You still haven't provided rationale for weighing liver? What about metabolism, energy partitioning?

  • This can be concluded based upon the results presented in the preceding sentences and measurements presented in table 2. Chicks with lower yolk weight had higher liver weight and also hatched first (presenting the time component). We have modified the wording to clarify our interpretation.

Line 257-260: In Introduction

  • These statements have been differentiated and an additional citation has been added to expand.

Line 289-291: Again, I would be more interested in your mortality data?

  • Please refer to section 3.2, lines 199-200, “Mortality was not affected (P > 0.05) by any of the treatments and the cumulative average was 2.57% at 42 d.”

Line 301-302: this could have been discussed more

  • This concept of environmental challenge, along with the current study, do suggest further studies would be beneficial. However, the authors respectfully request to only retain a brief mention of this in order to avoid speculation of these potential effects since they were not part of this study.

Line 316: References are over 10 years old, Lamont et al.,2014 is the most recent. Literature here is not extensive.

The authors believe your point stresses the importance of the current study. Relatively little research exists on the topic, particularly for modern broiler genotypes. As mentioned in the manuscript, much of the previous literature does not utilize the provision of nutrients in the hatch basket during hatch but instead involves fasted chicks after pull from hatcher. The differing experimental designs and conclusions as a result need to be differentiated by researchers. The newer “early feeding” articles most often investigate in ovo application, which results in different physiological responses and practical applications than the oral consumption of exogenous feed. Thus, these types of studies were not heavily referenced. While there are more recent review papers on the topic, there have been very few additions to the primary literature on this subject. One reference of relevance was added to the revised manuscript.